# High-Quality Dry Etching of LiNbO_3_ Assisted by Proton Substitution through H_2_-Plasma Surface Treatment

**DOI:** 10.3390/nano12162836

**Published:** 2022-08-18

**Authors:** Arjun Aryal, Isaac Stricklin, Mahmoud Behzadirad, Darren W. Branch, Aleem Siddiqui, Tito Busani

**Affiliations:** 1Center for High Technology Materials (CHTM), University of New Mexico, MSC01 04-2710, 1313 Godard St. SE, Albuquerque, NM 87106-4343, USA; 2Electrical and Computer Engineering (ECE), University of New Mexico, MSC01 11001, Albuquerque, NM 87131-0001, USA; 3Sandia National Laboratories, 1515 Eubank Blvd SE, Albuquerque, NM 87123, USA

**Keywords:** thin films, Lithium Niobate, microstructures, fabrication, plasma etching, silicon integration

## Abstract

The exceptional material properties of Lithium Niobate (LiNbO_3_) make it an excellent material platform for a wide range of RF, MEMS, phononic and photonic applications; however, nano-micro scale device concepts require high fidelity processing of LN films. Here, we reported a highly optimized processing methodology that achieves a deep etch with nearly vertical and smooth sidewalls. We demonstrated that Ti/Al/Cr stack works perfectly as a hard mask material during long plasma dry etching, where periodically pausing the etching and chemical cleaning between cycles were leveraged to avoid thermal effects and byproduct redeposition. To improve mask quality on X- and Y-cut substrates, a H_2_-plasma treatment was implemented to relieve surface tension by modifying the top surface atoms. Structures with etch depths as deep as 3.4 µm were obtained in our process across a range of crystallographic orientations with a smooth sidewall and perfect verticality on several crystallographic facets.

## 1. Introduction

Lithium Niobate (LiNbO_3_ or LN) has proven to be the material of choice for a wide range of applications due to its exceptional piezoelectric, electro-acoustical, electro-optical, and non-linear optical properties [1]. The different crystallographic orientations of LN are heavily utilized for applications in surface-acoustic-wave (SAW) resonators [2], optical filters [3], optical sensors [4], modulators [5,6,7], transducers [8,9], optical waveguides [10,11], Q-switch lasers [12,13], oscillators [14], etc. For example, high temperature (1000 °C) annealing Ti-doped X cut LN can be utilized as an optical waveguide [15]. At the nano- and micro-scale, device performance is often constrained by the fabrication quality of processed LN films [16]. However, unlike many semiconductors and dielectric materials, LN substrates are complex for processing and are notoriously difficult to etch [16]. Typically, a long plasma dry-etch is required to obtain high-aspect ratio or deep etching profiles [17] in LN substrates, and, hence, a successful LN etching process needs to manage factors including substrate heating, redeposition of the etching byproducts and mask materials, and durability of the mask materials over the etching process.

Dry etching of LN substrates was studied by different research groups [18,19,20,21,22,23,24,25,26] and different plasma etch conditions have been investigated to optimize the quality of the resulting structures. Fluorine (F_2_) based plasma sources have most commonly been used to etch LN [24]; however, the use of this source is accompanied by the redeposition of LiF and its byproducts, which reduce the etching rate and create unfavorable features due to local micro-masking effects. The byproducts from fluorine-based dry-etch evaporate at only 800 °C, thus remaining on the surfaces and the sidewalls of the structures. This not only reduces the etch rate of the target material, but also prevents obtaining vertical etching profiles [27]. Additionally, typical mask material delaminates or are otherwise consumed during the etching process, resulting in roughness on the etched structure, which can degrade performance (e.g., increased optical losses due to surface roughness). Adding Argon (Ar) gas to the chemistry of the plasma etching has been reported to enhance the physical sputtering process and obtain a more uniform etching profile [17], thus mitigating a portion of these challenges during the dry etch.

To overcome the problem of redeposition of product compounds, the lithium concentration on the top surface needs to be minimized. This can be achieved by employing a Proton Exchange (PE) process, where some of the lithium ions are replaced by protons in the LN crystal structure [28]. It has been shown that the PE process will significantly reduce the amount of LiF redeposition, and therefore increase the etching rate and improve the etching profiles as well [29]. Ti in diffused waveguides has been fabricated on Z cut LN. The proton exchange process was applied by immersing the substrate in benzoic acid at T = 240 °C for 3.5 h [30]. However, the PE process is expensive, not available for all processes and it is time consuming. Therefore, an alternative is needed to replace PE process with a cost-effective and a more available method.

Here, we report our recent progress in etching different crystalline LN substrates using Inductively Coupled Plasma (ICP). By comparing different combinations of the mask materials on different LN orientations (cuts), we present optimal dry etch conditions to obtain deep etching profiles on all orientations with smooth and vertical sidewalls. LN thin films are commonly used in phononic and photonic devices such as Surface Acoustic Waves (SAW), resonators, transducers, optical waveguides, optical filters, etc. To ensure the propagation of the acoustic and optical waves with minimum losses, a smooth and vertical etching profile is required. The scalloping, lateral etching and surface roughness can then severely affect the performance of the device [31,32]. Indeed, the Q-factor is directly impacted by the roughness [33,34,35].

As described by the Ruze’s formula in Equation (1) of the [35], the loss due to scattering can be described as, $\Delta G=-685.81\;{\left(\frac{\epsilon }{\lambda }\right)}^{2}\left(\mathrm{dB}\right)$. In other words, $Q\;\sim \;\pi \frac{\epsilon }{\lambda }$, where ϵ is the surface roughness and λ is the wavelength. Roughness at the hard edges of acoustic (or optical) devices can limit device Q-factors [32] and overall device performance. This effect is further aggravated as the frequency is scaled and can be a bottleneck for device performance. We also demonstrate that employing H_2_-plasma treatment prior to the processing step, as an alternative for the PE process, overcomes delamination of the metal mask on X- and Y-cuts LN samples and improves the quality of the etching profile.

## 2. Experiments and Results

A standard cleaning procedure: such as piranha clean (H2S04 and H202) followed by Acetone, IPA, DI water, and N_2_ dry air blow was carried out prior to the lithography process [36]. Then, a negative photoresist (AZnLOF 2035) was applied to pattern the LN samples with different cuts through a contact lithography process. We utilized single crystal LN substrates with common orientations of 128YX, 128Y, 64Y, and thin film LN on Si substrates with the orientations of X, and Y. These different types of LN wafers were supplied from NANOLN vendor. After patterning the samples, they were cleaned in an acid solution (HCl:H_2_O, 1:3) and metallized in a metal evaporator. The metals mask deposition was carried out around 1 × $10^{-6}$ torr without heating the substrate (nominal room temperature < 60 °C).

The whole fabrication procedure is schematically shown in Figure 1a. A hard mask consisting of Titanium (Ti), Aluminum (Al), and Chromium (Cr) was deposited. The process was followed by lift-off using 1165 (n-methyl pyrrolidinone) solution. Then, the samples were etched in a F_2_-based in Inductively Coupled Plasma (ICP). After etching, metal masks were removed using a diluted HF solution. Scanning Electron Microscopy (SEM) was applied to inspect and analyze the samples through the processing steps. Figure 1b,c show top and side view SEM image of the patterned photoresist after the lithography step, respectively. As seen, a perfect vertical sidewall was achieved to ensure formation of a high-quality metal mask after metallization process.

Identifying the right metal material for the hard mask for the etching purpose is a crucial step for metal depositions. Some dry etch challenges, such as redeposition of the mask material and the reaction products, can be avoided using the right metal for the processing. These challenges are more severe if the process needs to be designed to achieve deep structures on LN substrates, as are desired for optical and photonic device applications.

To find an optimum metal mask for a deep dry etch on LN samples, we tried different metal combinations; Ti/Al (20 nm/1.3 µm), Ti/Al/Cr (20 nm/750 nm/750 nm). Figure 2 compares the quality of the metallization on different LN cuts. As seen in Figure 2, the Ti/Al stack works perfectly on all types of LN cuts, while Ti/Al/Cr did not work perfectly on X and Y cut LN on the Si substrate, as shown in Figure 3; peeling off of the metal film was observed at the edges of the microstructures. In order to explain this observation, we hypothesize that both crystal orientation and the stress due to the transferring LN film on Si substrate play an important role in atomic bonding between metals and top surface of LN. Therefore, a surface treatment seems necessary to improve the quality of the Ti/Al/Cr metal mask on X- and Y-cuts.

A similar treatment was carried out by [24] to engineer crystal properties of the LN using proton implantation. Alternatively, the PE process can also be employed with molten benzoic acid. Quenching effects were studied on H*x*Li_1−_*x*NbO_3_ layers on Z cut LN waveguides. The hydrogen-doped *x* ∼ 0.47 in (001) LiNbO_3_ layers were created by proton exchange reaction in melt benzoic acid [26]. Here, we employed H_2_-Plasma (H2: 30 sccm, pressure: 15 mTorr, RF power: 150 W, ICP power: 300 W, DC bias: 53 V) in an inductively coupled plasma (ICP) chamber prior to the lithography and metallization steps in order to imitate the effect of proton treatment on the LN substrate. Figure 4 shows the metal mask quality on X- and Y-cuts after H_2_-plasma treatment. As seen, a remarkable improvement is obtained using plasma treatment.

After obtaining a high-quality metal mask on LN cuts, we proceeded to plasma dry etch using F_2_-based chemistry ICP. Previous research on etching LN revealed that formed Niobium Fluoride (NbF_5_) becomes volatile at temperatures around 200 °C, while the LiF byproducts will remain on the sidewalls and the surfaces that decreases the etching rate of the LN surfaces [25]. In our experiment, we used CHF_3_/Ar plasma treatment (CHF_3_: 30 sccm, Ar: 30 sccm, pressure: 5 mTorr, RF power: 100 W, ICP power: 150 W, DC bias: 40 V) to achieve high selectivity and higher etch rate with a Cr mask. The Argon gas increased the etching rate by enhancing the sputtering process during plasma etching [24]. To prevent heating effect on etching process (due to low thermal conductivity of LN), we implemented a periodic step process; 20 min etching followed by 4 min cooling (plasma is off). Then, after each 1 h of etching, the samples were cleaned in standard cleaning solvent (Acetone, IPA, DI water, and N_2_ dry air blow) Table 1 provides information on etching rates of different orientations of LN.

In Figure 5, we present the results using the described etching process for (a) the Ti/Al mask and (b) the Ti/Al/Cr mask. The presence of Al exposure to the plasma results in re-deposition of Al by-products, which are visible in Figure 5a as “pyramids” or “cones”. In order to prevent the re-deposition, we protected the Al with a layer of Cr. CrF_4_ by-products are more volatile compared to AlF_3_ [37,38], so that the formation of cones or pyramids is prevented. It is clear that the etching profile with Ti/Al/Cr metals worked as a better mask, since the surface roughness and redeposition underneath the sidewalls are prevented and, also, the verticality is nearly maintained, as illustrated in Figure 5b. Unfortunately, we were not able to achieve successful results using the Ti/Cr mask. Ti is necessary for ensuring adhesion between LN and the Al, while Al is needed to ensure adhesion with Cr. The thickness of the Al also seems to be a critical parameter, which is still under study. We think that Al, thanks to its electrical conductivity and thermal conductivity, prevents the accumulation of electrons during the etching process.

As observed in Figure 6, the right-angle verticality is nearly reached for the metal masks defined on a microstructure. There is only ~3° offset for the etched LN with the stacks of metal masks. The offset might have arisen from the lateral etching occurring during the plasma dry etching process. So far, our results are satisfying in terms of overall etching profile considering verticality (Figure 6) and redeposition free sidewalls. Indeed, other research groups tried to etch lithium niobate material for ridge and rib structures, using fluorine dry etching. This typically shows less verticality, between 60° and 78°, with a visibly rough side wall [39,40]. Unfortunately, the literature on LN tends to not report the quantitative values of the smoothness of the side walls. In Table 2, we present the roughness in terms of root mean square (rms) and arithmetic average, applying ImageJ software to our SEM images. For the samples presented in Figure 6b, the rms is ~10, while for the samples presented in Figure 7, the rms is an order of magnitude lower (~1.5). Based on image analysis and qualitative comparison with published SEM images of side walls, we can affirm that we achieved state-of-the-art smooth side walls and the most vertical profiles.

Table 1 lists the etch rates and the obtained depths for all the cuts using CF_3_/Ar (1:1) plasma dry etch. The etching rate may vary depending on the crystal orientation that may include properties of local neighborhood atoms, and role of the dangling and surface bonds. As seen in Table 1, the etching rates are nearly the same, but 64Y and 128Y cuts present a slightly lowering etch rate compared to the other cuts. Our etching rate results correspond similarly to the findings (Table 1, Ref. [20]), with minimum backside cooling of 10 sscm He flow. The etch rate with CHF3/Ar chemistry for PE bulk LN was reported to be 650 nm/h, which is close to our etch rate for bulk LN.

Table 2 represents the roughness measurements for different crystallographic orientations on the etched structures with arbitrary units. The roughness values were calculated using the ImageJ version 1.53s, image processing program created by Wayne Rasband and other contributors in the National Institute of Health, USA. This is the Fiji implementation of ImageJ in combination with the SurfChaJ plugin [41], which roughly calculates the roughness of a surface based on an image provided. Comparing the results, 64Y bulk LN is etched with minimum surface roughness compared to other bulk LN and thin films on Si.

## 3. Conclusions

In summary, we have shown an optimum ICP dry etch process to obtain a deep etching profile in LN substrate with minimum roughness and vertical sidewalls. The combination of CHF_3_/Ar (1:1) gases was used to etch all LN substrates. The plasma etching characteristics of different orientations of LN samples were studied under various metal masks. The stack of Ti/Al/Cr metals was found to be the best mask material for all cuts, i.e., 128YX, 128Y, 64Y, X-, and Y-cuts, as it removed all the redeposition and micro-masking issues during the dry etch process. However, to achieve the same quality of metal mask, X- and Y-cuts samples were treated with H_2_-Plasma. It was assumed that the plasma treatment will mitigate surface stress on the transferred LN film due to the Si substrate and also modify the surface structures by replacing Li-ion with protons. Likewise, we included periodic interruption steps in our plasma dry etch method, followed by chemical cleaning between each cycle to avoid thermal effect and minimize byproduct redeposition during the long etching process. Microstructures with etch depths as deep as 3.4 µm were achieved using this method with a smooth sidewall and perfect verticality, which is promising for the fabrication of optical devices, where high-aspect ratio structures are required.

## Figures and Tables

**Figure 1 nanomaterials-12-02836-f001:**
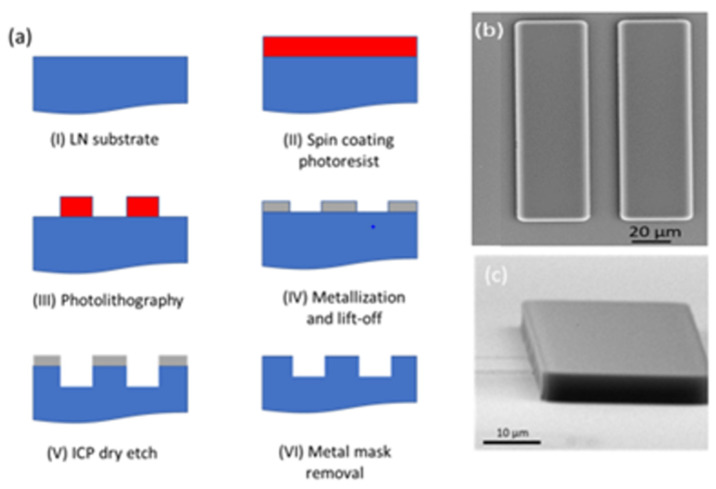
(**a**) Schematic of the fabrication process to make pattern on LN samples. (**b**) Top view SEM image of a sample after lithography. (**c**) Side view SEM image of patterned photoresist after lithography.

**Figure 2 nanomaterials-12-02836-f002:**
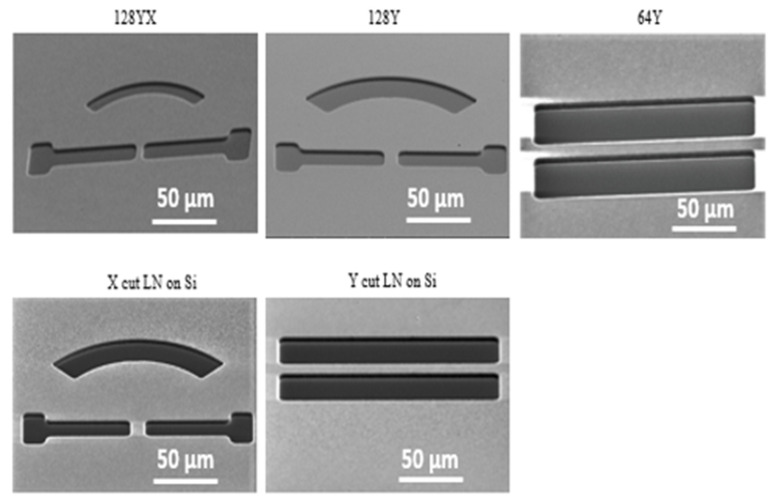
Comparison of Ti/Al mask quality on different LN cuts.

**Figure 3 nanomaterials-12-02836-f003:**
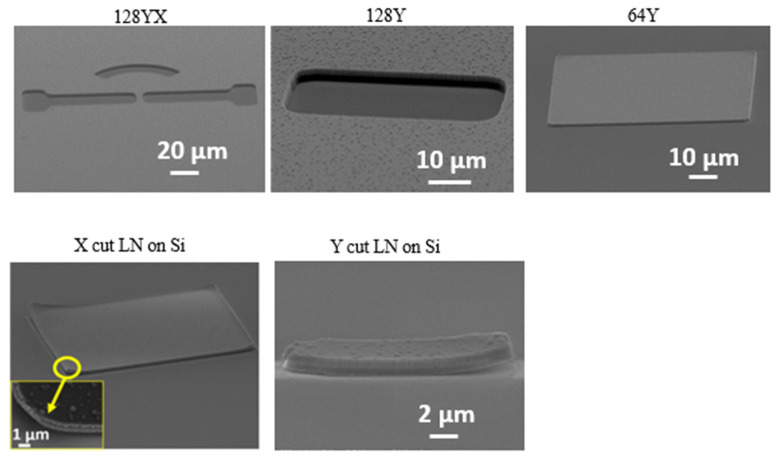
Comparison of Ti/Al/Cr mask quality on different LN cuts. Inset shows peeling off the metal mask X- and Y-cuts at the edge of the structures.

**Figure 4 nanomaterials-12-02836-f004:**
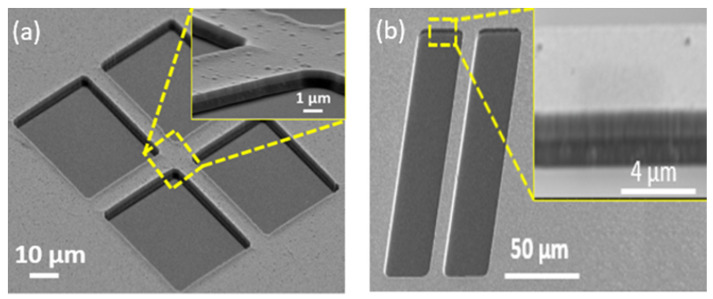
Metal mask quality on (**a**) X-cut and (**b**) Y-cut LN after employing H2-plasma treatment.

**Figure 5 nanomaterials-12-02836-f005:**
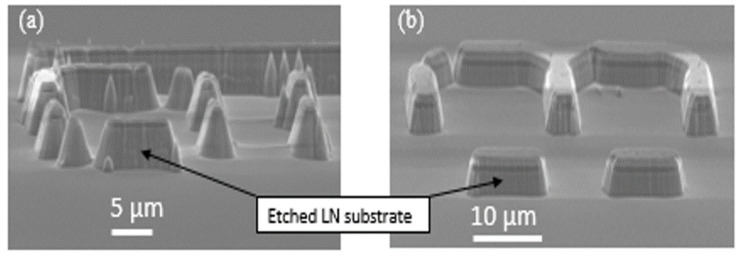
128YX LN etching using CHF_3_/Ar and metal mask of (**a**) Ti/Al and (**b**) Ti/Al/Cr.

**Figure 6 nanomaterials-12-02836-f006:**
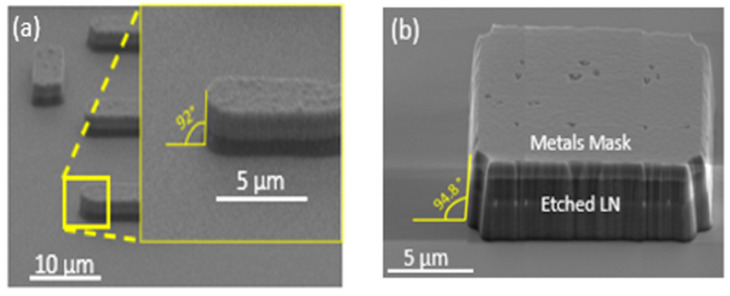
Side wall features on LN (**a**) Ti/Al/Cr metal mask and (**b**) 128Y LN etched using CHF_3_/Ar gas.

**Figure 7 nanomaterials-12-02836-f007:**
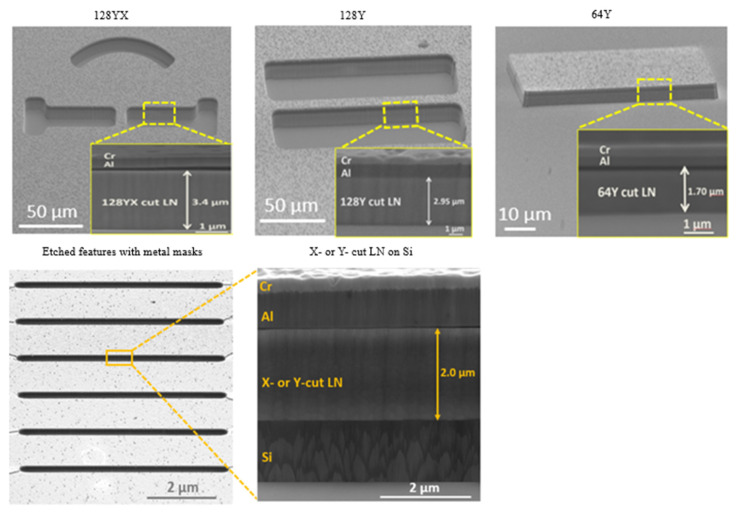
ICP Etching of a micro-structure using CHF_3_/Ar gases with Ti/Al/Cr metal masks.

**Table 1 nanomaterials-12-02836-t001:** Etching rates of different cuts of LN using the CHF_3_/Ar plasma.

Orientation	Etching Depth (μm)	Etch Rate of LN (μm/h)
** 128YX (Bulk LN) **	3.40	0.70
** 128Y (Bulk LN) **	2.95	0.65
** 64Y (Bulk LN) **	1.70	0.55
** X with H+ treatment (Thin Film) **	2.00	0.70
** Y with H+ treatment (Thin Film) **	2.00	0.75

**Table 2 nanomaterials-12-02836-t002:** Roughness Measurements (Arbitrary Units).

	128YX	128Y	64Y	X- or Y Cut Thin LN Film on Si
$R_{q}$ (**Root mean square roughness**)	1.669	9.511	1.273	10.262
$R_{a}$ (**Arithmetic average roughness**)	1.274	7.496	1.044	9.254

## Data Availability

The data reported in this manuscript are available on request from the corresponding author.

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
