# Peer review of "High-Quality Dry Etching of LiNbO3 Assisted by Proton Substitution through H2-Plasma Surface Treatment"

_nanomaterials, 2022, doi:10.3390/nano12162836_

Round 1

Reviewer 1 Report

This is an high scientific quality work presenting an otimized etching methodology to obtain high quality etch profiles in  hard to etch LiNbO3 films.

The research methodology was clearly though and scientific and conclusions is clearly presented. I undoubtedly recommend this work for publication.

Author Response

We thanks the reviewer for the time spent during the review process. We went through spell check and correct some typos and grammar errors.

The title also has changed as

‘’High Quality Dry Etching of LiNbO3 assisted by proton substitution through H2-Plasma Surface Treatment’’

Reviewer 2 Report

The manuscript demonstrates an optimized microprocessing method to obtain a nearly vertical smooth etching of LN films. The results are interesting and will be useful for many important applications. The manuscript can be considered for acceptance by Nanomaterials if the following comments are substantially addressed.

1. The key points for achieving the vertical smooth etching were not clear. It is necessary to provide some scientific discussions on the possible mechanism of vertical smooth etching and the possible contributions such as mask materials and etching parameters.

2. Only optimum results were provided in the manuscript, it is also necessary to show the etching results using the conventional methods for comparison.

3. Line 85, Shematically should be schematically.

4. In Figure 3, the scale bars in all figures are too small to be identified, please make necessary revisions.

5. Please measure the roughness of the sidewall of the etched structures, which is vital data for proving the significance of the work.

6. In the top panel of Figure 7, the scale bars in all insets are also not clear, please make some improvements.

7. The manuscript does not include any data or application of Scalable RF MEMS of LN films declared in the title.

8. The funding information is missing.

Author Response

We would like to thank you for the valuable comments and suggestions that will certainly improve the quality of the paper. All the comments and suggestions has been revised and updated accordingly.

Regards

Reviewer 3 Report

My comments are in attachement.

Author Response

We would like to thank you for your careful reading of the manuscript. Our responses to your comments are updated and revised. The corresponding line numbers are indicated for each of the comments and suggestions.

We believe that the manuscript, in the revised form, addresses your concern.

Thanks to some comments provided by another reviewer, we decided to change the title as follow:

‘’High Quality Dry Etching of LiNbO3 assisted by proton substitution through H2-Plasma Surface Treatment’’

Round 2

Reviewer 2 Report

All my comments have been addressed adequently. I suggest the current version can be considered for acceptance by Nanomaterials.